# Predictors of virological outcomes after analytical interruption of antiretroviral therapy and HTI vaccination in early treated people with HIV-1.

Lucia Bailón [1,2,17], Yovaninna Alarcón-Soto [1,17], José R. Arribas[3,4,5], Adrian Curran[6], Anuska Llano[7], Samandhy Cedeño [8], Roger Paredes [1,3,8,9,10,11], Elena Vendrame [12], Devi SenGupta[12], Jeffrey J. Wallin[12], Romas Geleziunas[12], Marie P. Malice[13], Ian McGowan[14,15], Christian Brander [3,8,10,14,16,17] ✉, José Moltó[1,3,17] ✉ & Beatriz Mothe [1,3,8,10,17]

## Abstract

**Background** Randomized, placebo-controlled clinical trials (RCTs) that include analytical treatment interruptions (ATI) are conducted to test the efficacy of HIV cure strategies. Two independent RCTs, AELIX-002 and AELIX-003, evaluated the HTI immunogen-based vaccines alone or combined with the TLR7 agonist vesatolimod in early-treated people with HIV (etPWH). These studies individually demonstrated that higher levels of vaccine-induced HTI-specific T-cell responses were associated with extended time off antiretroviral therapy (ART) during a 24-week ATI.

**Methods** We conducted a pooled analysis of both RCTs including the individual data of a total of 88 participants. The association between clinical, immunogenicity and viral data and rebound outcomes during the ATI was evaluated using logistic regression and receiver operating characteristic (ROC) analyses.

**Results** We identify an HTI-specific threshold of 835 spot-forming cells/$10^6$ peripheral blood mononuclear cells as a predictor of delayed and slower viral rebound during ATI. This threshold distinguishes participants who remain off ART for >12 weeks, with 58% sensitivity, 85% specificity, 75% positive and 73% negative predictive value.

**Conclusions** These findings confirm that HTI-specific T-cell magnitude at ATI initiation is the strongest predictor of ATI outcomes observed in AELIX002/003 studies and that a threshold of vaccine-induced HTI-specific T-cell responses can be used as futility criteria before ATI and/or guide participant selection in future HTI-based HIV cure trials aimed at achieving therapy-free remission.

## Plain language summary

People with HIV have a weakened immune system and so cannot respond well to other infections. Current treatment usually requires people to take antiretroviral therapy (ART) for life. Vaccination could potentially help the body's immune system control the virus without daily treatment. To test how well vaccinations work, people with HIV were asked stop taking ART. In this study, we combined the results from two clinical trials that tested different combinations of HIV vaccines that were given alone or with another drug that stimulated the immune system. We found that participants who developed higher levels of immune responses to the vaccines were more likely to keep the virus under control for longer periods of time after stopping ART. Importantly, we found an amount of vaccine response that could predict which participants could remain withoutART for at least 12 weeks. This information could be useful in future HIV treatment studies to recommend which study participants can or cannot stop ART, which is helpful to reduce the risks associated to ART interruption.

Over the past decade, interest has grown in developing immune strategies to induce durable HIV-1 control without antiretroviral therapy (ART). However, no validated biomarkers currently predict virological control after ART cessation, leaving analytical treatment interruptions (ATI) and

inclusion of placebo groups as the only reliable tools to assess the efficacy of therapeutic interventions[1,2]. Published guidelines describe safe conduct of ATI in clinical trials, ensuring that participants can re-suppress HIV-1 plasma viral load (pVL) upon resuming ART with no/minimal increases in

viral diversity or reservoir size[3–5]. Nonetheless, ATI poses concerns for participants and their sexual partners due to potential risk of transmitting HIV[6–8]. Additionally, including placebo groups in clinical trials is debatable, as it significantly increases trial costs and complexity, and impacts participant acceptability. Although recently published data could serve as a reference to refine sample size calculations and minimize the number of participants undergoing ATI in future HIV remission trials, the identification of a biomarker capable of predicting viral outcomes would help avoid unnecessary ATI in cases with a low likelihood of success.

AELIX-002 (NCT03204617) and AELIX-003 (NCT04364035) were independent, randomized, placebo-controlled clinical trials primarily designed to assess the safety of HIVACAT T-cell immunogen (HTI)-based vaccines either alone or combined with the Toll-like receptor-7 (TLR7) agonist vesatolimod in early-treated people with HIV. The main objectives on safety, immunogenicity and efficacy of both studies have been published previously[9,10]. Secondary objectives included evaluating T-cell immunogenicity and viral rebound during a 24-week ATI. Both trials demonstrated the induction of strong, broad and HTI-focused T-cell responses associated with ATI outcomes, but limited sample sizes precluded establishing a predictive cutoff of the HTI-specific T-cell magnitude for extended ART-free periods. Therefore, we conducted a post-hoc pooled analysis of AELIX-002 and AELIX-003 to identify predictors of ATI outcomes. In this analysis, we identify an immune correlate that, with adequate specificity and sensitivity, predict several virological outcomes during ATI and that may help discern which individuals are more likely to remain off therapy for longer periods in future clinical trials.

## Methods
### Data source
This analysis included individual data from participants in the AELIX-002 (NCT03204617) and AELIX-003 (NCT04364035) clinical trials who underwent an ATI. Study populations consisted of individuals who initiated ART within 6 months of the estimated HIV-1 acquisition date and maintained an undetectable viral load while on ART. Participants in AELIX-002 received a combination of DNA.HTI, MVA.HTI and ChAdOx1.HTI vaccines, while those in AELIX-003 received ChAdOx1.HTI and MVA.HTI in conjunction with the TLR7 agonist vesatolimod, or matched placebo (Supplementary Fig. 1). Both trials included a 24-week ATI with weekly monitoring of pVL, starting at 8 and 12 weeks of the last MVA.HTI boost in AELIX-002 and AELIX-003, respectively. ATIs in both trials used identical ART resumption criteria, including a single HIV-1 pVL ≥100,000 copies/ml, 8 consecutive weeks with >10,000 copies/ml, two consecutive CD4 + T-cell counts <350 cells/mm³ and/or a reported grade ≥3 acute retroviral syndrome. Demographic and clinical characteristics were recorded at study entry. HIV-1 reservoir and immunogenicity parameters were measured at inclusion and at ATI start, before ART was interrupted. Total and intact proviral (IPDA) HIV-1 DNA copies in CD4⁺ T cells were quantified in extracts of lysed CD4⁺ T cells using digital droplet PCR processed at Accelevir Diagnostics. Total HTI and HIV-1-specific T cells were assessed ex vivo using isolated PBMCs with an IFN-γ-detecting enzyme-linked immunoabsorbent spot assay (ELISpot IFN-γ Mabtech kit), as previously described[9]. Briefly, 15-mer peptides overlapping by 11 amino acids were combined into 10 peptide pools (7–22 peptides per pool) spanning different HIV-1 proteins/sub-proteins the HTI vaccine insert (P1–P10) and eight pools of 62–105 peptides per pool spanning the rest of the HIV-1 viral protein sequences (OUT P1–P8). Magnitude of the HTI-specific T-cell response was defined as the sum of SFCs per 10⁶ PBMCs for HTI pools (P1–P10). The breadth of the total HTI-specific T-cell response was defined as the number of reactive peptide pools out of the 10 HTIs covering peptides pools used in the ELISpot assay.

AELIX-002 and AELIX-003 were conducted in compliance with the ethical principles of the Declaration of Helsinki, ICH harmonized tripartite guideline E6(R2). Both studies were approved by the Ethics Committee of Hospital Universitari Germans Trias i Pujol, Badalona, Spain (IRB00002131) and the Spanish regulatory (AEMPS) agency. Written

informed consent in compliance with regulatory authority regulations was obtained from all participants. Detailed descriptions of the AELIX-002 and AELIX-003 trials can be found elsewhere[9,10].

### Statistics and reproducibility
No power calculation was conducted for this analysis, as the sample size included individual data from all participants who entered an ATI in AELIX-002 and AELIX-003. Time to viral load rebound was calculated from the ATI start date to the date of first occurrence of at least two consecutive determinations of pVL ≥50 copies/ml. Time off ART was calculated from the ATI start date to the date of ART resumption. The time to event was derived using the calendar date of the event minus the ATI start date +1 and then transformed to weeks. The Kaplan–Meier estimator was used to describe the time to ART resumption, and survival functions were compared using the Gehan–Breslow–Wilcoxon test. Differences in medians between groups were compared using the Mann–Whitney and Fisher tests, when corresponding. Spearman's $\rho$ and Benjamini–Hochberg were used for correlations unadjusted or adjusted for multiple comparisons, respectively. All tests were two-sided, with 5% error rate. To identify individual characteristics that could influence the binary outcome of time off ART ≥ 12 weeks versus <12 weeks, post-hoc univariate logistic regression models were used. Covariates with a $p$-value < 0.25 in the univariate analysis were selected for inclusion in the multivariate logistic models following an assessment for multicollinearity and excluding covariates that were highly correlated among them. A backward stepwise regression approach was then used to retain only the statistically significant covariates. Subsequently, to estimate the discriminatory performance of variables associated with time off ART > 12 weeks in the multivariate analysis, a ROC curve analysis was performed, and 95% CI was computed with 2,000 stratified bootstrap replicates. Analyses were performed by using R project 4.2.2 (https://www.r-project.org/) and GraphPad Prism version 10.4.1 for Windows (GraphPad Software, https://www.graphpad.com).

## Results
### Study population
Data from 88 virologically suppressed individuals (recipients: vaccine, 55 men, 1 woman; placebo, 32 men) who underwent ATI in AELIX-002/003 trials (Supplementary Fig. 1a, b) were pooled. Both study populations had comparable demographic and clinical characteristics (Supplementary Table 1). Levels of intact (but not total) HIV-1 DNA at ATI start were numerically lower in AELIX-002 versus AELIX-003, possibly due to greater use of INSTI-based ART as first-line therapy[11]. As described[9,10], HTI vaccination elicited high HTI-specific T-cell frequencies but did not impact HIV proviral DNA.

### ATI outcomes
During ATI, pVL was monitored weekly and ART resumed[9,10] after a single HIV-1 pVL ≥100,000 copies/ml, eight consecutive pVL ≥10,000 copies/ml and/or two consecutive CD4 counts <350 cells/mm³. All participants experienced HIV-1 rebound (pVL >50 copies/ml) after ART discontinuation, with comparable kinetics across studies (Supplementary Fig. 1c, d). Overall, 28% (25/88) participants remained off ART for 24 weeks.

### Correlates analysis
Clinical characteristics, viral reservoir, and immune parameters at study entry and ATI start were assessed for associations with ATI outcomes, both overall and in the active and placebo groups separately (Fig. 1). Lower pre-ART viremia and longer time on ART were associated with delayed viral rebound, longer time off ART and lower HIV-1 pVL at end of ATI. Similarly, lower levels of total and intact HIV-1 proviral DNA both at study entry and at ATI start correlated with slower virus recrudescence (time to pVL >10,000 copies/ml), extended time off ART and lower pVL at end of ATI. Notably, higher magnitude and breadth of HTI-specific T-cell responses at ATI start were significantly associated with all ATI outcomes in vaccine recipients, including delayed viral rebound, slower recrudescence and

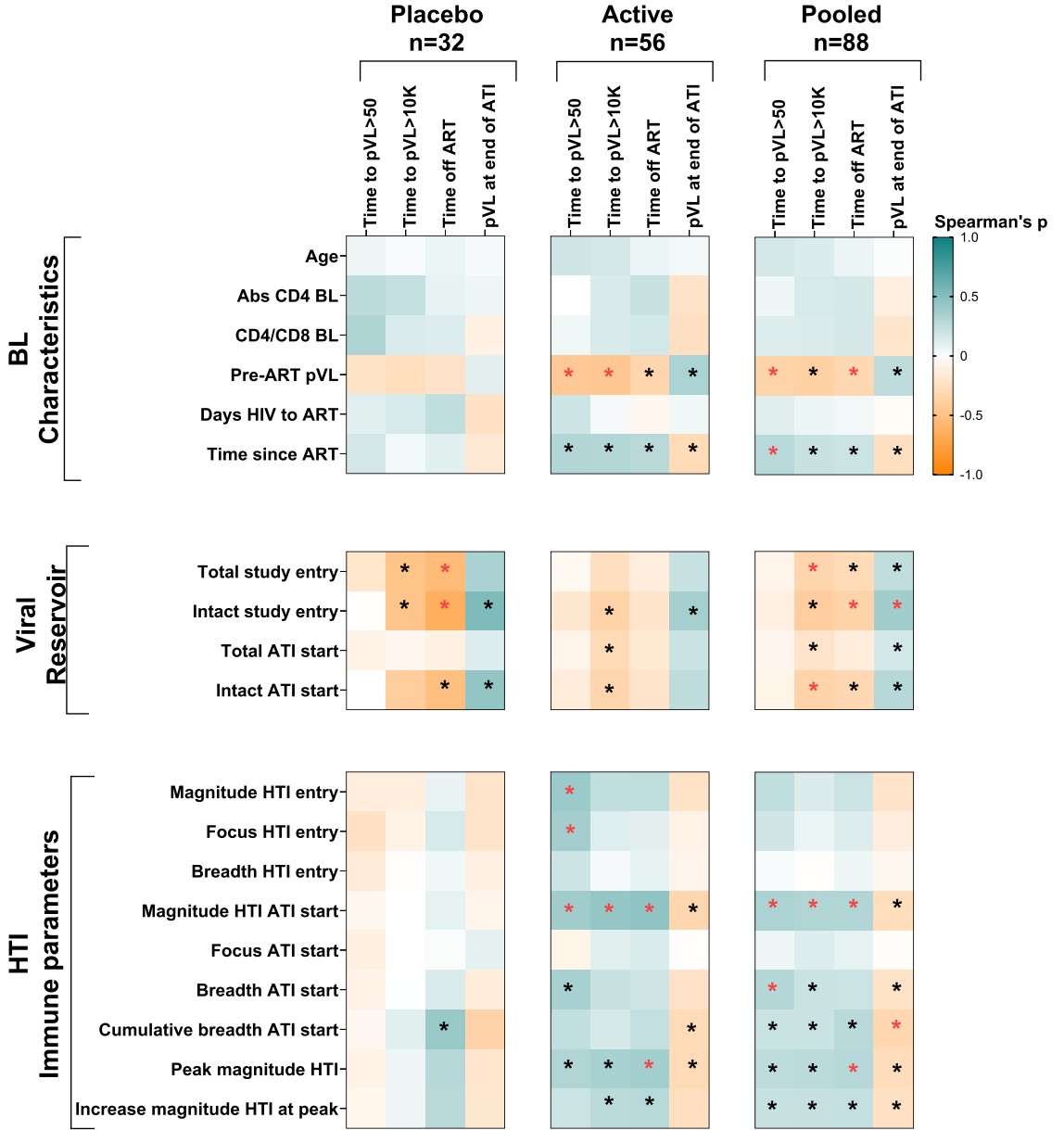

**Fig. 1 | Clinical, virological and immunological correlates of ATI outcomes.**
Correlogram for active, placebo and pooled data for all AELIX-002 and AELIX-003 participants. Spearman's ρ is used for correlations. Tests are two-sided, unadjusted for multiple comparisons, with 5% error rate. Exact ρ and p-values can be found in Supplementary Table 2. Asterisks denote significant correlations when *P* < 0.05, in red those remaining significant when adjusted for multiple comparisons by Benjamini–Hochberg test. The numerical source data underlying Fig. 1 is provided in Supplementary Table 2.

prolonged time off ART (*P* < 0.05 for all, Supplementary Table 2). Of note, after adjusting for multiple testing comparison, HTI magnitude at ATI start was the variable more consistently associated with better ATI outcomes (Fig. 1).

Given the binary distribution of time off ART (≤12 or >12 weeks, one of the pre-specified secondary endpoints in both AELIX-002/003 trials), logistic regression analysis was used to define predictors of remaining off ART for >12 weeks (*n* = 36) versus ≤12 weeks (*n* = 52). After assessing multicollinearity and excluding covariates that were highly correlated among them, covariates with *P* < 0.25 in the univariate models were selected for multivariate analysis (Supplementary Table 3). Higher magnitude of HTI-specific T cells and lower total proviral HIV-1 DNA at ATI initiation were independently associated with remaining off ART > 12 weeks, with odds ratios (95% CI) of 1.13 (1.05–1.23) for each 100 HTI-specific spot-forming cells (SFCs)/$10^6$ peripheral blood mononuclear cells (PBMCs)

increase, and 0.79 (0.62–0.95) for each 100-copy increase in total HIV-1 DNA/$10^6$ CD4 T cells, respectively (Supplementary Table 4).

To test whether HIV-1 proviral DNA levels and HTI immunogenicity at ATI start predicted time off ART > 12 weeks, we conducted receiver operating characteristic (ROC) curve analyses. Total HIV-1 DNA had limited predictive value (AUC [95% CI] 0.61 [0.49–0.74]) (Fig. 2a), compared to HTI magnitude ≥835 SFCs/$10^6$ PBMCs, which showed a moderate predictive performance to distinguish participants remaining off ART > 12 weeks (AUC [95% CI] 0.68 [0.55–0.81]) (Fig. 2b). This cutoff value, which corresponded to the median of the observed HTI magnitude across active arms, demonstrated 58% sensitivity, 85% specificity, 75% positive predictive value (PPV) and 73% negative predictive value (NPV) for the entire cohort of participants. Considering vaccinees only, the sensitivity of this threshold increased to 83%, specificity to 76%, and PPV and NPV to 73% and 85%, respectively. Vaccine recipients with HTI-specific magnitude

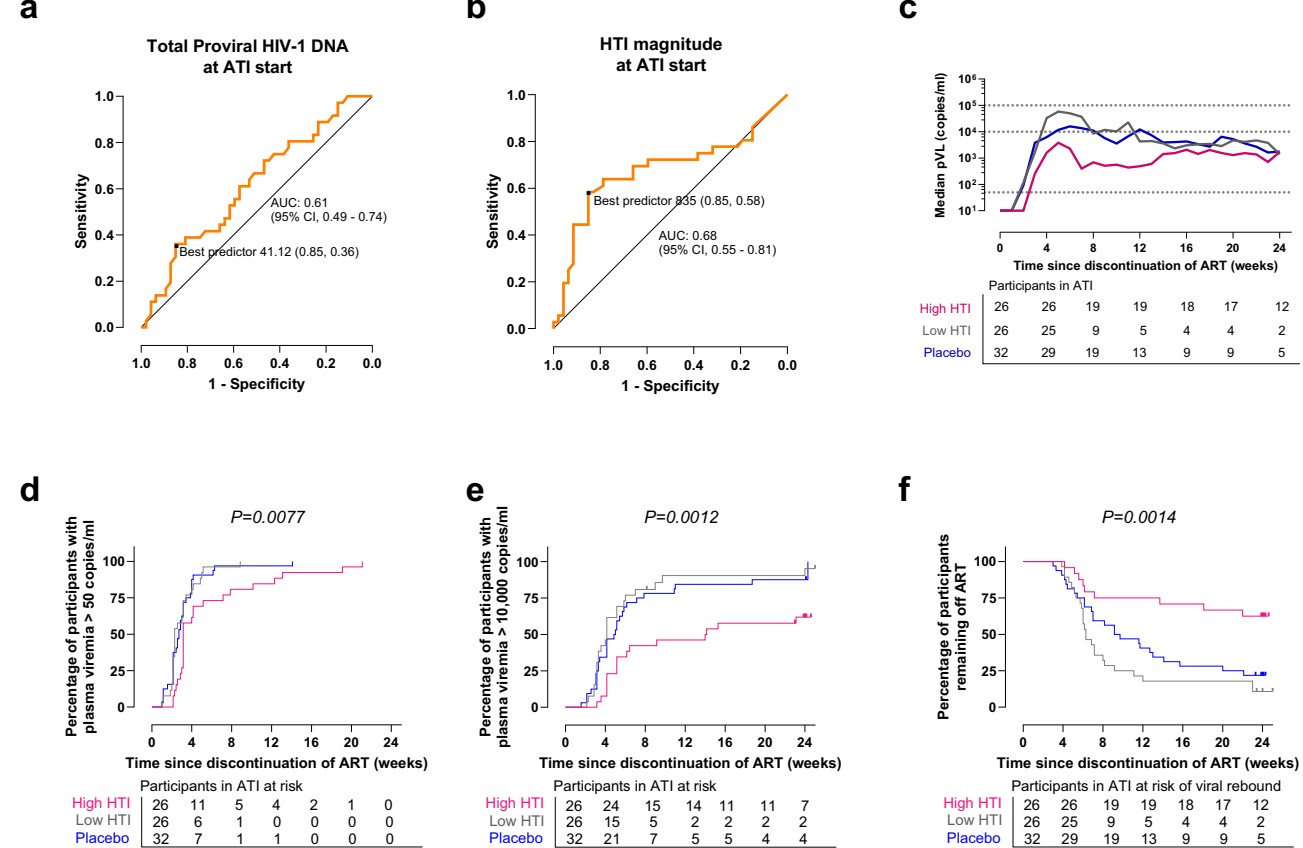

**Fig. 2 | HIV-1 reservoir and immunogenicity threshold.** ROC curves for **a** total proviral HIV-1 DNA at ATI start and **b** magnitude of HTI-specific T-cell response at ATI start. AUC (95% CI) and best predictor (specificity, sensitivity) are shown. **c** Median pVL during ATI, **d** time to pVL >50, **e** >10,000 copies/ml and **f** time off ART in vaccine recipients with HTI magnitude above (high HTI) or below (low HTI) 835 SFCs/$10^6$ PBMCs at ATI start, and for placebo recipients. Gehan-Breslow-Wilcoxon test is used. Tests are two-sided, unadjusted for multiple comparisons, with 5% error rate.

at ATI start ≥835 SFCs/$10^6$ PBMCs had significantly delayed and slower viral rebound and longer time off ART versus those below this threshold (Fig. 2c), whose ATI outcomes were comparable to placebo recipients (Gehan-Breslow-Wilcoxon test, $P = 0.0077$, $P = 0.0012$ and $P = 0.0014$, respectively, Fig. 2d–f). Median (IQR) time off ART was 23.9 (6.7–24.1) weeks for vaccine recipients with HTI magnitude ≥835 SFCs/$10^6$ PBMCs at ATI start, versus 6.3 (2.3–9.6) weeks for vaccinees with HTI magnitude <835 SFCs/$10^6$ PBMCs and 9.7 (6.1–22.1) weeks for placebo recipients, only 6% of whom had HTI magnitude ≥835 SFCs/$10^6$ PBMCs at ATI start.

## Discussion

HIV remission trials face challenges, particularly regarding ATI and inclusion of placebo arms. Identification of correlates of ATI viral control is urgently needed as such biomarkers could help prioritize promising combination strategies, accelerate HIV cure research and mitigate risks of failure at later stages of development. Additionally, these biomarkers could be incorporated into trial designs as futility criteria preceding the ATI, preventing unnecessary ATI-related risks for individuals unlikely to control viral rebound below ART restart criteria. However, small sample sizes, variability in trial populations and ATI designs, and limited efficacy signals have hindered biomarker identification. Here, we pooled individual data from two independent trials testing HTI vaccines in comparable populations using the same ATI design, and in which a modest efficacy signal provided the basis for biomarker discovery. Therefore, the key added value of this pooled analysis lies in the increased sample size, which allows for the identification of a potential cutoff of HTI-specific T cell responses that could predict extended time off ART during ATI and that could be used to inform the design of future HIV remission studies.

While HTI vaccines induced strong cytotoxic T-cell responses that contribute to viral control, they did not reduce the HIV reservoir or prevent viral rebound. Thus, combining them with immunomodulators, B-cell vaccines, and/or broadly neutralizing antibodies remains an important area of investigation. Regardless of regimen, reliable biomarkers predicting ATI outcomes will be key for future studies.

Our findings suggest HTI-specific T-cell magnitude at ATI initiation, measured in cryopreserved PBMCs using an IFN-γ ELISpot assay, is the strongest predictor of ATI outcomes observed in AELIX002/003 studies. A threshold of 835 HTI-specific SFCs/$10^6$ PBMCs predicted extended ART-free time (>12 weeks) with acceptable accuracy. Notably, vaccinees with low HTI responses exhibited similar viral dynamics to placebo recipients, of whom only rare exceptions exceeded the HTI magnitude threshold. According to our results approximately 3 out of 4 individuals predicted to remain off ART > 12 weeks would do so, while the same proportion of those predicted to resume ART < 12 weeks actually did. We acknowledge that the PPV and NPV may reflect just a moderate predictive performance. However, in the context of ATI in HIV-remission trials, these values are potentially meaningful to guide the design of future ATI trials by minimizing the number of participants interrupting ART that are unlikely to reach any certain degree of post-intervention control beyond 12 weeks of ATI. However, further refinement and external validation with data from future studies are warranted. Noteworthy, the 835 SFC/$10^6$ cut-off was also statistically associated with delayed viral rebound. Nonetheless, the absolute difference in time to 1st detectable pVL during the ATI was minimal between participants above or below the threshold, questioning its clinical value for implementation in ART-remission trial design.

We also found that lower levels of total HIV-1 DNA correlated with most ATI outcomes; however, since the 95% of the CI of the AUC in the ROC analysis included 0.5, we were not able to exclude the possibility that its predictive performance was no better than chance, precluding us to establish a reliable cutoff value for total HIV-1 DNA to predict extended time off ART. While the relationship between proviral DNA and clinical endpoints has been inconsistent across studies[12–14], in a recent pooled analysis of individual data from six interventional trials including an ATI, total HIV-1 DNA levels correlated with time to detectable viremia (pVL >50 copies/ml) and time to loss of viral control (pVL >5000 copies/ml) during ATI[15]. However, proviral DNA determination requires digital droplet PCR-based measurements, specialized laboratories and lengthy analysis, limiting its applicability as a futility marker during trial execution.

Our findings have implications for the design of future combination studies using HTI-based immunotherapies. For instance, the threshold of the HTI magnitude could be applied as a futility criterion before entering the ATI, while keeping the blinded nature of the study. Alternatively, in the case that the study design would recommend to unblind participants before the ATI start to prevent placebo recipients to undergo ATI or because of a cross-over design[16] the immune correlate could be useful for vaccine recipients as could help to predict participants unlikely to control with higher accuracy. Therefore, using HTI magnitude as an ATI initiation criterion could reserve ATI for participants more likely to achieve viral control, and their rebound dynamics could be compared with available data from nonintervention cohorts[17]. Moreover, while placebo groups remain valuable for safety and immunogenicity assessments, avoiding unnecessary ATI in placebo recipients and vaccinees unlikely to control viremia would enhance trial acceptability and address several ethical, logistical and economic challenges in current HIV remission research[18]. This approach aligns with a recent report that used machine learning and mathematical modeling to propose decision rules within the first weeks of ATI to identify predictive biomarkers for ATI post-treatment control and recommend earlier ART resumption in those less likely to control[19].

Several considerations should be noted when interpreting our results. First, the AELIX-002/003 populations were restricted to early-treated people with HIV, with lack of representation of cis-gender and transgender women, potentially limiting generalizability to individuals starting ART during later stages of HIV and more diverse populations. In these cases, HTI-induced responses might differ, and larger and more escaped reservoirs could impact viral control upon ART interruption. Second, our findings are specific to the HTI vaccines studied here and may not be generalizable to other therapeutic T-cell vaccines Therefore, further research is necessary to validate our results for other vaccine candidates including or not HTI-regions in their immunogen constructs. Finally, we used logistic regression models and ROC analysis to identify predictors of time off ART > 12 weeks. Although this timepoint might be seem arbitrary, it reflects the observed distribution of time off ART in our studies, using consensus ART resumption criteria, and is also supported by the most recent meta-analysis on pVL rebound kinetics[15]. This meta-analysis confirmed that sustained control of pVL <50 copies/ml beyond 12 weeks is rare, and that most individuals have already achieved a post-ATI pVL setpoint by this timepoint. In conclusion, this pooled analysis of AELIX trials identified a cutoff value for the frequency of HTI-specific T cells at ATI initiation that predicts prolonged time off ART. These findings may directly contribute to optimizing the design of future studies testing HTI-based combination strategies aimed at achieving durable ART-free HIV control.

## Reporting summary

Further information on research design is available in the Nature Portfolio.

## Data availability

The numerical source data underlying Fig. 1 is provided in Supplementary Table 2. Gilead Sciences share anonymized individual participant data such as demographic, lab values, etc., as well as related documents such as study protocols, statistical analysis plans, upon request or as required by law or regulation with qualified external researchers based on submitted curriculum vitae and reflecting non conflict of interest. The request proposal must also include a statistician. Approval of such requests is dependent on the nature of the request, the merit of the research proposed, the availability of the data and the intended use of the data. The data will be available for 1 year from the signing of Gilead's Data Sharing Agreement contract with an option of 3-month extensions at the deadline if further time is needed for research. Data requests should be sent to datarequest@gilead.com, and requestors can expect a response within 3 business days. For further information regarding Gilead's Data Sharing Policy, please visit GileadClinicalTrials.com.

## Code availability

Access to the R code used for this specific analyses is available upon request or as required by law or regulation with qualified external researchers based on submitted curriculum vitae and reflecting non conflict of interest. The request proposal must also include a statistician. Approval of such requests is dependent on the nature of the request, the merit of the research proposed, the availability of the data and the intended use of the data. The code will be available for 1 year from the signing of Gilead's Data Sharing Agreement contract with an option of 3-month extensions at the deadline if further time is needed for research. Code requests should be sent to datarequest@gilead.com, and requestors can expect a response within 3 business days. For further information regarding Gilead's Data Sharing Policy, please visit GileadClinicalTrials.com.

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

## Acknowledgements
Special thanks to all the volunteers participating in these studies. Editing and production assistance for the manuscript were provided by Parexel and funded by Gilead Sciences, Inc. This study was funded by AELIX Therapeutics and Gilead Sciences, Inc, and supported by grants from the Spanish Research Agency ISCIII PI20/01039, the European Union's Horizon 2020 research and innovation program under grant agreement #101057548 (EPIVINF) and NIH grant P01- AI178375. L.B. held a PhD grant from ISCIII (RH CM20/00097) during the conduct of the study. The sponsors of the studies, AELIX Therapeutics and Gilead Sciences, Inc., contributed to the data analysis and interpretation, and provided a review of the manuscript.

## Author contributions
C.B., J.M. and B.M. conceived and designed the study. L.B. and Y.A.-S. additionally contributed to the study design and data analyses. L.B., J.R.A., A.C., J.M. and B.M. contributed with clinical development of the study. A.L., S.C., E.V., D.S., R.G., I.M., C.B., J.M. and B.M. contributed to data management and overall study coordination. Y.A.-S., M.P.M, J.J.W. and J.M. undertook the statistical analysis. L.B., J.M. and B.M. drafted the manuscript. Y.A.-S., J.R.A., A.C., R.P., E.V., D.S., R.G., I.M., C.B., J.M. and B.M. critically revised the manuscript for important intellectual content. All authors reviewed and approved the final version of the manuscript. The AELIX-002 and AELIX-003 Study Groups contributed to participant recruitment, clinical management of participants, and data collection. All members of the consortia reviewed and approved the final manuscript.

## Competing interests
L.B. reports speaker fees from Gilead Sciences and ViiV Healthcare, outside the scope of the submitted work. J.R.A. reports advisory and speaker fees and grant support from ViiV, Janssen, Gilead Sciences, MSD and AELIX Therapeutics. A.C. reports advisory and speaker fees and grant support from Gilead Sciences, Janssen, MSD and ViiV Healthcare. R.P. reports research grants from Merck Sharp & Dohme, ViiV Healthcare, Gilead Sciences and PharmaMar; consultancy fees for Gilead Sciences, Pfizer, Roche Therapeutics, AstraZeneca, Merck Sharp & Dohme, GSK, ViiV Healthcare, Eli Lilly and Company, and PharmaMar; and honoraria from Gilead Sciences, Pfizer, AstraZeneca, Merck Sharp & Dohme and PharmaMar, outside of the submitted work. E.V., D.S., J.J.W., and R.G. are employees and stockholders of Gilead Sciences. I.M. was an employee of AELIX Therapeutics S.L. at the time of the study, and is currently an employee of Orion Biotechnology, and is a consultant for Synklino, outside the scope of the submitted work. C.B., B.M., and A.L.L. are co-inventors of the HTI immunogen (patent application PCT/EP2013/051596). C.B., B.M., and I.M. are co-inventors of US patent application no. 62/935,519 and US patent application no. 62/851,546, which have relevance to the vaccine regimen used in both studies. C.B. was co-founder, Chief Scientific Officer, and shareholder of AELIX Therapeutics S.L. and serves as an advisor for Tendel Therapies and OmniScope, outside of the submitted work, and reports speaker fees from Gilead Sciences. J.M. has received research funding, consultancy fees and lecture sponsorships from, and has served on advisory boards for, various laboratories (MSD, AbbVie, Boehringer Ingelheim, Gilead Sciences, Viiv Healthcare, Janssen-Cilag and Bristol Myers Squibb). B.M. reports past consultancy personal fees from AELIX Therapeutics S.L., as well as speaker fees from Gilead Sciences, Janssen, MSD and ViiV Healthcare, outside the submitted work. The remaining authors declare no competing interests.

## Additional information

[1]Department of Infectious Diseases and Fundació Lluita contra les Infeccions, Hospital Universitari Germans Trias i Pujol, Badalona, Spain. [2]Department of Medicine, Autonomous University of Barcelona, Cerdanyola del Vallés, Spain. [3]CIBER de Enfermedades Infecciosas (CIBERINFEC), Instituto de Salud Carlos III, Madrid, Spain. [4]Department of Internal Medicine, Infectious Disease Unit, Hospital Universitario La Paz, Madrid, Spain. [5]Hospital La Paz Institute for Health Research (IdiPAZ), Madrid, Spain. [6]Infectious Diseases Department, Hospital Universitari Vall d'Hebron, Vall d'Hebron Research Institute (VHIR), Barcelona, Spain. [7]Unitat Secretaria Tècnica CEIm Parc Taulí. Parc Taulí Hospital Universitari. Institut d'Investigació i Innovació Parc Taulí (I3PT-CERCA), Sabadell, Spain. [8]IrsiCaixa, Hospital Universitari Germans Trias i Pujol, Badalona, Spain. [9]Universitat Politècnica de Catalunya – BarcelonaTech, Terrassa, Spain. [10]Department of Pathology, Center for Global Health and Diseases, Case Western Reserve University School of Medicine, Cleveland, OH, USA. [11]Department of Infectious Diseases and Immunity, University of Vic-Central University of Catalonia, Vic, Spain. [12]Gilead Sciences, Inc, Foster City, CA, USA. [13]StatAdvice, Inc, Brussels, Belgium. [14]AELIX Therapeutics S.L, Barcelona, Spain. [15]Division of Gastroenterology, Hepatology, and Nutrition, Department of Medicine, University of Pittsburgh, Pittsburgh, PA, USA. [16]Catalan Institution for Research and Advanced Studies (ICREA), Barcelona, Spain. [17]These authors contributed equally: Lucia Bailón, Yovaninna Alarcón-Soto, Christian Brander, José Moltó, Beatriz Mothe. ✉e-mail: cbrander@irsicaixa.es; jmolto@lluita.org

