## [Transparent Peer Review file · Communications Medicine]

Predictors of virological outcomes after analytical interruption of antiretroviral therapy and HTI vaccination in early treated people with HIV-1.

Corresponding Author: Dr Beatriz Mothe

Version 0:

Reviewer comments:

Reviewer #1

(Remarks to the Author)

I've been asked to comment on the revisions performed to Reviewer #3's concerns. The responses are reasonable. However, though this short form article makes it quite hard to sufficiently explain some of the nuanced statistical analyses so I would not be surprised if some other readers also struggle with some of the points R3 made (especially the <12 >12 binary, the multiregression with exclusions for collinearity). I appreciate the statistical rigor of the present work but I do find it leaves me hoping for a more intuitive answer to the question, does HTI level correlate with PTC? Most of us would be looking for a "dose-response" with some intuitive feature of rebound.

Just a few minor other points:

In their response, the authors write "HTI magnitude as a futility criterion before entering the ATI, while keeping the blinded nature of the study, or even, considering unblinding participants before the ATI." I thought that was a thought-provoking point that could be articulated in the paper

Related to that, overall these results would benefit from honest and careful wording about how HTI specific responses (and your cutoff) would translate to other vaccine/immunotherapeutic responses

Fig1 is ok but it would be better to see which of these associations actually survive multiple testing comparison -- less likely to be mis-interpreted by less statistically fluent readers

Fig2c would benefit from more obvious notation that those lines are medians taken across 3 groups

the authors may want to be slightly more careful with calling "reservoir", and for instance in l176: "size determination requires..." might be more precise to say intact proviral DNA measurements require (as what the "reservoir" actually is is somewhat still a manner of opinion)

09 October 2025

On behalf of the co-authors, thank you for additional comments to the manuscript **“Predictors of virological outcomes after analytical interruption of antiretroviral therapy following immune interventions with HTI vaccines in early-treated people with HIV: a pooled analysis from the AELIX-002 and AELIX-003 clinical trials”**.

We would like to thank reviewer #4 for additional comments on our revised submission.

Please find below a table with the author point by point responses to the specific reviewer and locations of the corresponding updates in the new version of the manuscript, together with the revised versions of the documents (with and without “track changes”).

Yours sincerely,

Dr. Christian Brander

Dr. José Moltó

Dr. Beatriz Mothe

Reviewer	Comment number	Comments	Authors response	Location of changes
Reviewer 4	1	I've been asked to comment on the revisions performed to Reviewer #3's concerns. The responses are reasonable. However, though this short form article makes it quite hard to sufficiently explain some of the nuanced statistical analyses so I would not be surprised if some other readers also struggle with some of the points R3 made (especially the <12 >12 binary, the multiregression with exclusions for collinearity). I appreciate the statistical rigor of the present work but I do find it leaves me hoping for a more intuitive answer to the question, does HTI level correlate with PTC? Most of us would be looking for a "dose-response" with some intuitive feature of rebound.	We thank the reviewer to comment on revisions performed by Reviewer #3. The answer to the question: does HTI level correlate with PTC? Is yes, as described in the abstract (lines 50-51), and in lines 186-188 "Vaccine recipients with HTI-specific magnitude at ATI start ≥ 835 SFCs/10^6 PBMCs had significantly delayed and slower viral rebound and longer time off ART versus those below this threshold" We have tried to explain the multicollinearity assessment in a clearer way for the readers, specifying that variables showing collinearity were excluded for the multivariate logistic models. We would like to emphasize that when multicollinearity occurs, it indeed makes it difficult to determine the independent effect of each variable on the response variable, leading to unstable or even incorrectly assigned regression coefficients, inflated standard errors, and unreliable p-values. To minimize such risks, we excluded covariates that were highly correlated among them. This approach follows standard recommendations (Daoud, J. I. (2017). Multicollinearity and regression analysis. Journal of Physics: Conference Series, 949, 012009).	Lines 189-197
	2	In their response, the authors write "HTI magnitude as a futility criterion before entering the ATI, while keeping the blinded nature of the study, or even, considering unblinding participants before the ATI." I thought that was a thought-provoking point that could be articulated in the paper	Following this comment, we have included this discussion in the revised version of the manuscript.	Lines 271-272
	3	Related to that, overall these results would benefit from honest and careful wording about how HTI specific responses (and your cutoff) would translate to other vaccine/immunotherapeutic responses	As noted by the reviewer, our findings are specific to the HTI vaccines and may not be generalizable to other therapeutic T-cell vaccines. Therefore, further research is	Line 291-294.

			necessary to validate our results for other vaccine candidates including or not HTI-regions in their immunogen constructs. This limitation is described in the manuscript in the discussion section	
4	Fig1 is ok but it would be better to see which of these associations actually survive multiple testing comparison -- less likely to be mis-interpreted by less statistically fluent readers		After adjusting for multiple testing comparison, HTI magnitude at ATI start was the variable more consistently related to ATI outcomes. This information has been included in the revised version of the manuscript. We have kept the original figure but added a footnote in the figure legend to highlight which parameters remain statistically significant after multiple comparisons adjustment.	Lines 189-191 Figure 1
5	Fig2c would benefit from more obvious notation that those lines are medians taken across 3 groups		Following reviewer's comment, we have specified that lines in Figure 2c are medians across the 3 groups.	Figure 2
6	the authors may want to be slightly more careful with calling "reservoir", and for instance in l176: "size determination requires..." might be more precise to say intact proviral DNA measurements require (as what the "reservoir" actually is is somewhat still a manner of opinion)		We thank the reviewer for this comment. We totally agree with it and we have changed the wording accordingly throughout the manuscript	Lines 172, 203,259,263,267.